# Enhancing Water-Sensitive Urban Design in Chiang Mai through a Research–Design Collaboration

Chulalux Wanitchayapaisit [1,2], Nadchawan Charoenlertthanakit [1], Vipavee Surinseng [1], Ekachai Yaipimol [1], Damrongsak Rinchumphu [2] and Pongsakorn Suppakittpaisarn [1,*]

1   Department of Plant and Soil Sciences, Faculty of Agriculture, Chiang Mai University,
    Chiang Mai 50200, Thailand; chulalux.w@cmu.ac.th (C.W.); mrsnadchawan.c@cmu.ac.th (N.C.);
    vipavee.s@cmu.ac.th (V.S.); ekachai.y@cmu.ac.th (E.Y.)
2   Faculty of Engineering, Chiang Mai University, Chiang Mai 50200, Thailand; damrongsak.r@cmu.ac.th
*   Correspondence: pongsakorn.sup@cmu.ac.th

**Abstract:** Water-sensitive urban design (WSUD) is a subset of nature-based solutions (NbSs) that are implemented worldwide. However, the WSUD guidelines in some local contexts, such as Southeast Asia, remain unclear both for ecological and cultural reasons. This study aims to gather collaborations between researchers, designers, and laypeople in WSUD, which have the potential to be implemented to address water quality issues. The study consisted of three stages: site selection, a design workshop, and public interviews. Utilizing geo-design principles and geographical data, the potential pilot site was identified: a vacant space next to the Tha Phae Gate Plaza. A two-day workshop with landscape design experts yielded six conceptual designs, focusing on diverse themes such as water treatment, plant-based solutions, educational opportunities, and cultural enrichment. Public interviews provided insights into the community's perspectives on stormwater management, desired amenities, environmental considerations, and governance concerns. The results highlighted a collective interest in using NbSs for stormwater treatment and enhancing the area's recreational and educational potential. This study offers a comprehensive approach to addressing water quality issues in urban settings while considering local cultural, recreational, and environmental needs.

**Keywords:** stormwater management; green infrastructure; research-through-design method

## 1. Introduction

Urban areas are rapidly expanding. Consequently, impervious surfaces are covering the land [1–3]. When rainwater cannot be absorbed, it can cause floods and damage to riverbanks and can carry pollutants into clean water sources [4,5]. These pollutants can make water unsafe for plants, animals, and humans [6,7]. To tackle this issue, experts around the world are using natural methods, such as gardens designed to absorb rainwater. Water-sensitive urban design (WSUD) is one such approach. It strategically uses natural elements to treat and manage water in urban spaces [8]. One example of the elements in WSUD is rain gardens, which are gardens designed with underground layers to address stormwater issues [9,10]. However, to make these solutions work, there needs to be a stronger partnership between the people who study these problems and those who design our cities [11]. This article aims to explore how these designs can be introduced in places like Chiang Mai and how the public might perceive them.

### 1.1. Urban Stormwater Issues

Water management and land use for reducing environmental impacts have become global priorities, especially in urban spaces [12]. Urban areas are experiencing rapid growth, resulting in the reduction in natural surfaces [13]. Structures like buildings and roads are essential for an increasing population, but their expansion contributes to decreased permeability, which interrupts the water cycle [10,14]. These impermeable surfaces could

cause the surge in water to be too fast for natural absorption. This could lead to the destruction of natural riverbanks and water bodies and flooding in urban areas [10,14]. Furthermore, heavy rainfall can carry urban pollutants and chemicals into natural water sources, leading to water-related issues and a surge in nutrient levels in water bodies [15,16].

These nutrients, primarily nitrogen and phosphorus [17] from human activities like fertilizers and urban waste, trigger eutrophication. In large quantities, these issues compromise water oxygen levels, rendering the water unsuitable for various plant and animal species that thrive in pure water. This situation poses a risk to human health [18].

Addressing these challenges requires exploring ways to enhance infiltration rates on existing land surfaces. Such strategies are called nature-based solutions (NbSs) [19,20]. Across the globe, the strategies are also adapted under different names like water-sensitive urban design (WSUD) in Australia, low-impact development (LID) and green stormwater infrastructure (GSI) in the United States, and active, beautiful, clean waters (ABC Waters) in Singapore. These strategies provide valuable approaches for improving soil surfaces for higher infiltration rates [21].

### 1.2. WSUD: A Potential Solution

In countries addressing water management challenges, innovative approaches like water-sensitive urban design (WSUD) have become increasingly popular in recent years [22]. These methods utilize plants to aid in water treatment, a process known as phytoremediation [23], which offers a promising solution for urban eutrophication issues. Eutrophication occurs when water bodies accumulate excessive nutrients, causing problems like algae blooms and disrupting aquatic ecosystems [24,25]. Phytoremediation employs carefully selected plants that can absorb and reduce these nutrient concentrations, which can mitigate these issues in urban areas [23].

WSUD is a comprehensive concept and methodology dedicated to urban water management, emphasizing efficient water resource usage and environmental impact reduction. Integrated within WSUD are various sustainable water management approaches. This combination provides a sophisticated strategy for managing water in urban areas, which focuses on controlling the hydrological cycle and enhancing water quality through localized facilities [8].

The key principles of WSUD are collecting, retaining, and infiltrating surface runoff. This approach not only helps in retaining stormwater but also plays a vital role in recharging groundwater and replenishing soil water [8]. Furthermore, it contributes to preserving organic layers beneath the ground's surface [26].

WSUD employs several different strategies, including 'rain gardens', along with other elements aligned with the green stormwater infrastructure (GSI) principle [27]. Rain gardens (Figure 1), along with other GSIs, use plants and other natural elements to manage stormwater [9]. They are widely recognized and adopted worldwide as tangible illustrations of WSUD principles, providing functional stormwater management while enhancing the aesthetic and cultural aspects of urban landscapes. This aligns with the broader objectives of sustainable and water-sensitive urban development [28]. Furthermore, it provides additional ecosystem services such as thermal comfort, wildlife habitat, and human health and well-being [29–31].

### 1.3. Rain Gardens Designs, Instructions, and Constraints

Because of their potential benefits within WSUD, principles, guides, and suggestions for implementing rain gardens have appeared in urban planning and stormwater management guides across the world in the past decade [8,9,26,32–34]. The key anatomy of rain gardens includes two main parts: hardscape and softscape [10]. The hardscape includes layers like the soil layer, soil porosity, and ponding depth [9], and the softscapes include plants, either trees, shrubs, or groundcovers [32].

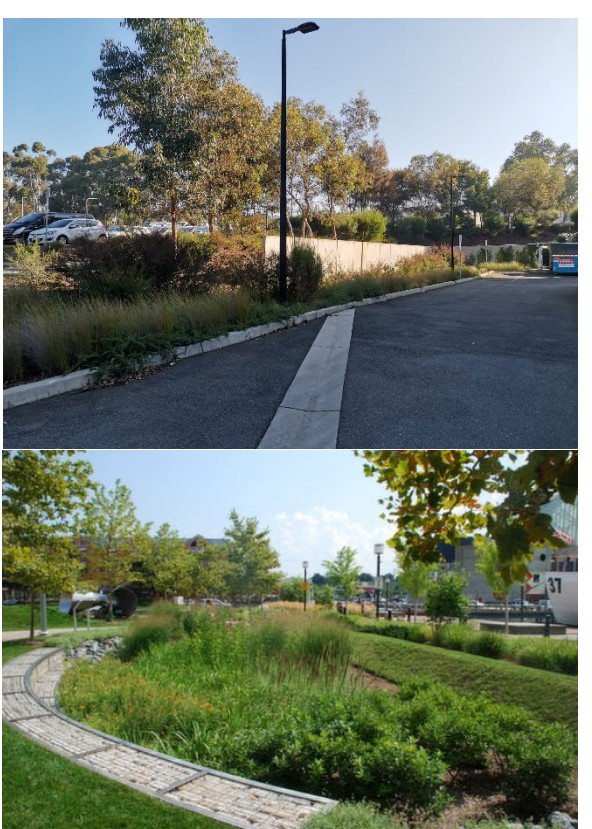
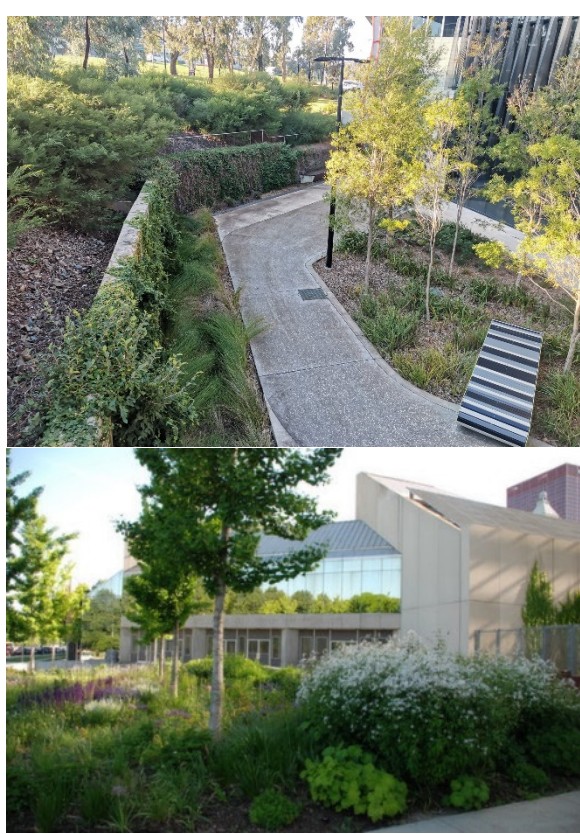

**Figure 1.** Examples of rain gardens across Australia (**top**) and the United States (**bottom**). Images from Pongsakorn Suppakittpaisarn and Xiangrong Jiang.

Referencing the Technical Guidance Manual widely adopted in several U.S. states [35,36], an ideal location is an area that is downslope of the stormwater catchment area or the impervious area from which stormwater tends to originate such as a driveway or rooftop [37]. There are a variety of designs available for hardscapes. For example, Australian guidelines [38] prescribe layer specifications: a ponding depth layer (20–30 cm), filter media layer (30 cm), transition layer (10 cm), and storage layer (30–40 cm). The depths of these layers are contingent upon the precipitation levels in each specific area. In regions experiencing high rainfall, the depth of these layers will be more extensive and deeper to accommodate the increased volume of precipitation [10]. Materials include sand and sandy loam for the filter media, sand for transition, and large gravel for storage.

In contrast, the knowledge on softscape designs is highly limited. Existing data primarily feature plant species from temperate zones [39,40]. Insights into plant species for hot and humid places, like Thailand, remain limited, with principles drawn from Singapore [41,42]. However, a comprehensive literature review underscores the significant potential of plants in water absorption. Rainwater discharges via surface run-off or infiltration through fissures and pores and disperses through the soil where plants are rooted. The complex root systems play an important role in facilitating the infiltration of rainwater into the soil [43]. Moreover, the transpiration process depends on the plant morphology and stomatal behavior. If plants receive proper influence, they will transpire large amounts of water. This can speed up the drying process of soil after a storm event. This results in the soil's ability to retain water increasing [44]. Nevertheless, there is a limited set of clear examples and principles guiding WSUD design in Thailand or Southeast Asia, especially concerning suitable rain garden designs.

*1.4. Gaps between Researchers, Designers, and Laypeople*

Aside from the lack of clear technical guides, there needs to be stronger connections between researchers, designers, and laypeople in landscape architecture, design, and planning. In 2000, a team of researchers interviewed a group of landscape architects and found that while academia had shifted to a focus towards sustainability, practitioners opted to design mainly for human uses [45]. Similar studies in 2017, which analyzed the speech contents of practitioners, showed a positive trend towards more empirical evidence in landscape architecture and design [46]. However, gaps are often expressed between researchers and designers in the field [47,48]. Traditionally, practitioners expected researchers to produce the answers, and the researchers provided something that explains the questions rather than practical solutions [47,49]. Thus, more modern trends in landscape architecture, design, and planning research have started focusing on the collaboration of the inquiries and started to blend research and design together in various ways, including studio-based research and research-through-design methods [47,48]. However, these solutions are under development and need examples and standards to illustrate what the processes might look like and what precautions may be needed in such collaborations.

Thus, this study posed a question: what might the process and results of collaboration between designers and researchers in creating a WSUD element prototype, such as a rain garden, for Chiang Mai look like, and how might they be received by the public?

## 2. Materials and Methods

*2.1. Overview*

Our study seeks to address water quality issues in Chiang Mai using water-sensitive urban design (WSUD) with a focus on eutrophication stemming from non-point source pollution. We selected the research-through-design framework [48,50] as the key method of this study. The study unfolds in three phases: site selection, design workshop, and on-site interviews.

*2.2. Site Selection*

2.2.1. Chiang Mai City Moat as the Focal Point

We chose Chiang Mai City Moat as our primary area of site selection. Chiang Mai City Moat is an area of historic Chiang Mai. The moat is an artificial canal surrounding approximately 1 square mile of the old city, a tourist destination [51,52]. The city moat's water is connected to the Ping River, which releases into the Chao Phraya River running past central Thailand into the Andaman gulf. In modern times, the moat has been used more as blue infrastructure for the aesthetic and thermal comfort of the city and is a part of Chiang Mai's attraction [52]. In addition, locals and homeless people have also used it as a source of food, fishing and diving for freshwater shellfish. The city moat has experienced eutrophication during the dry season in recent years, resulting in unpleasant odors and risking negative experiences among tourists [53]. Chiang Mai Municipality has installed oxygen suppliers into the water and has been planting trees along the moat, but it needs further applications to prevent the problem.

Although the water volume from the moat is small compared with the remaining Ping River, an on-site WSUD demonstration at the city moat can draw tourist attention and serve as a case study for other interested parties. Figure 2 locates Chiang Mai City Moat within the city of Chiang Mai.

LEGENDS

- ▢ Low-density residential area
- ▢ Mid-density residential area
- ▢ High-density residential area
- ▢ Commercial area
- ▢ Thai arts and culture preserves
- ▢ Nature preserves
- ▢ Recreational area
- ▢ Educational area
- ▢ Religious area
- ▢ Governmental institution

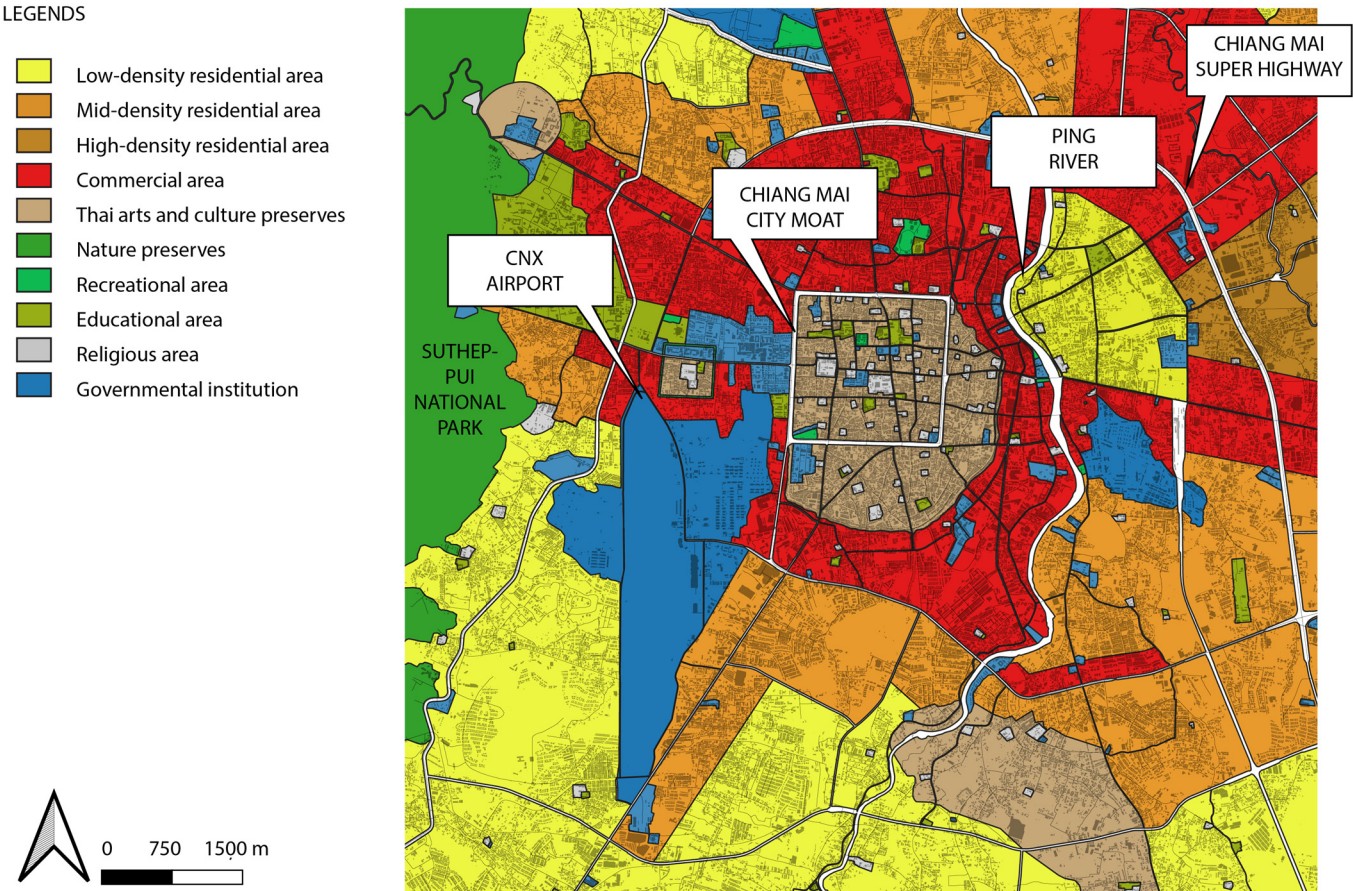

**Figure 2.** Chiang Mai City Moat within the context of the current city. Map dataset provided by Assoc. Prof. Manat Srivanit from Dhammasat University. Planning colors adapted from the Thailand's Department of Public Works and Town and Country Planning color-coding [54].

2.2.2. Site Selection for Pilot Demonstration

We used geo-design frameworks to start the pilot site selection process. This process is an accepted framework for landscape architecture and planning implementations [55]. First, we used the map data of the city of Chiang Mai collected by Dhammasat University in 2022. We considered the availability of land use (green spaces, temples, governmental offices, and educational institutions,) site median elevation, median slope, proximity to popular tourist sites, low tree cover density, drainage directions towards the city moat, and proximity to the moat. The analysis was completed using QGIS 3.6 software. We found a series of suitable sites for our pilot demonstration, as shown in Figure 3. Noticeably, the sites with higher suitability scores were concentrated around the Tha Phae Gate of the city moat.

Once the suitability data were gathered, we approached Chiang Mai Municipality for their opinions. The officers considered the data and suggested an underutilized site next to the Tha Phae Gate.

Tha Phae Gate (the gate of marinas,) originally called Chiang Ruek Gate (the gate of the boating district,) was the westernmost gate and the closest city gate to the Ping River. It is the gate that connects the royal temple to the river. The original form of the gate was lost, but it was reconstructed in the year 1984 in reference to the main gate of San Fang Temple [51]. The large plaza surrounding the gate was one of the most iconic spaces in the city. Tourists gathered there for photoshoots, outdoor concerts, and weekend markets [56].

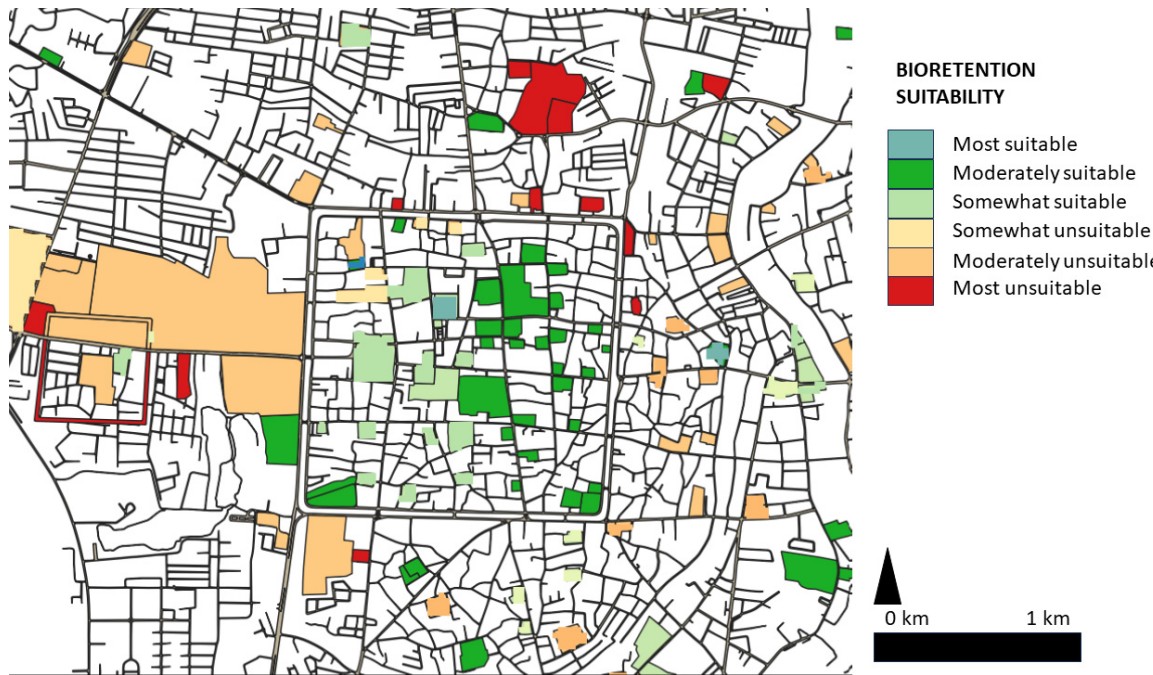

**Figure 3.** Suitable sites around Chiang Mai City Moat.

The main site, however, is adjacent to the plaza but connected by a narrow alley behind the shops. It is a circular semi-plaza of about 1200 square meters that releases its runoff right into the moat. The site was originally a roundabout for vehicles, but it was blocked off to formal vehicular access due to traffic issues. It is currently under-utilized and often misused for unauthorized parking or occupied by homeless residents. The underutilization of this space was perceived as a missed opportunity and a risk of perceived safety by the municipality despite its potential. Figure 4 displays the urban space of Tha Phae Gate and the site, while Figure 5 displays the floor plan of the site.

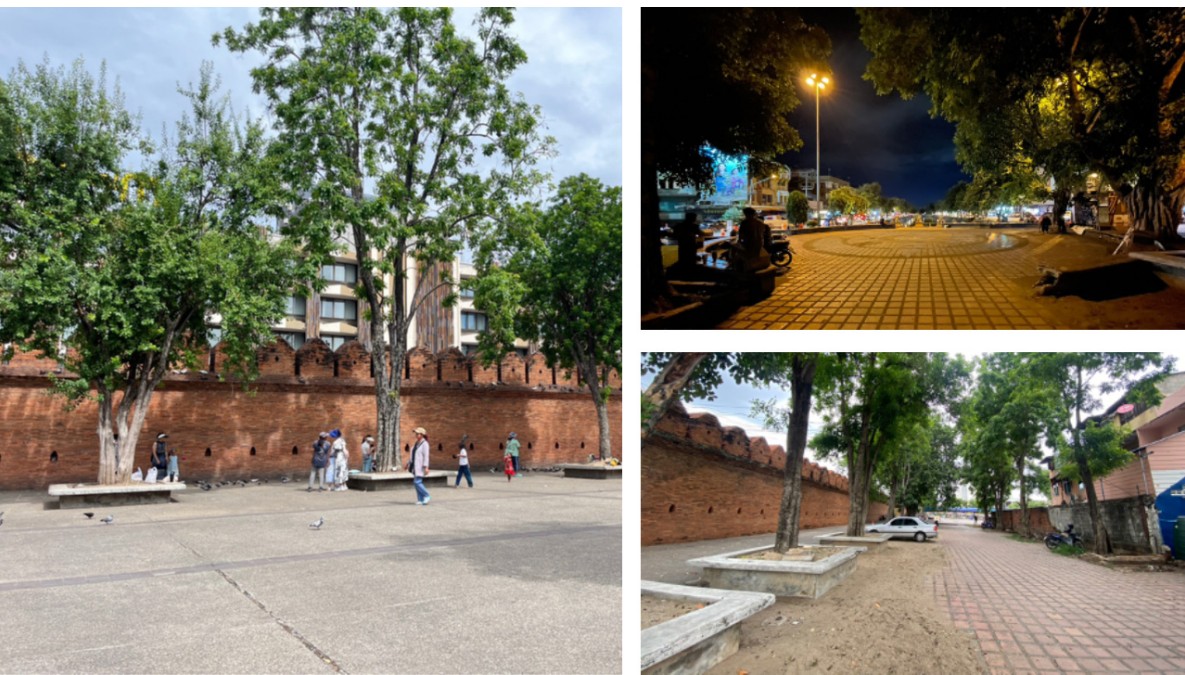

**Figure 4.** Tha Phae Gate Plaza (**left**), the site (**top right**), and the alley connecting the site to the main plaza (**bottom right**).

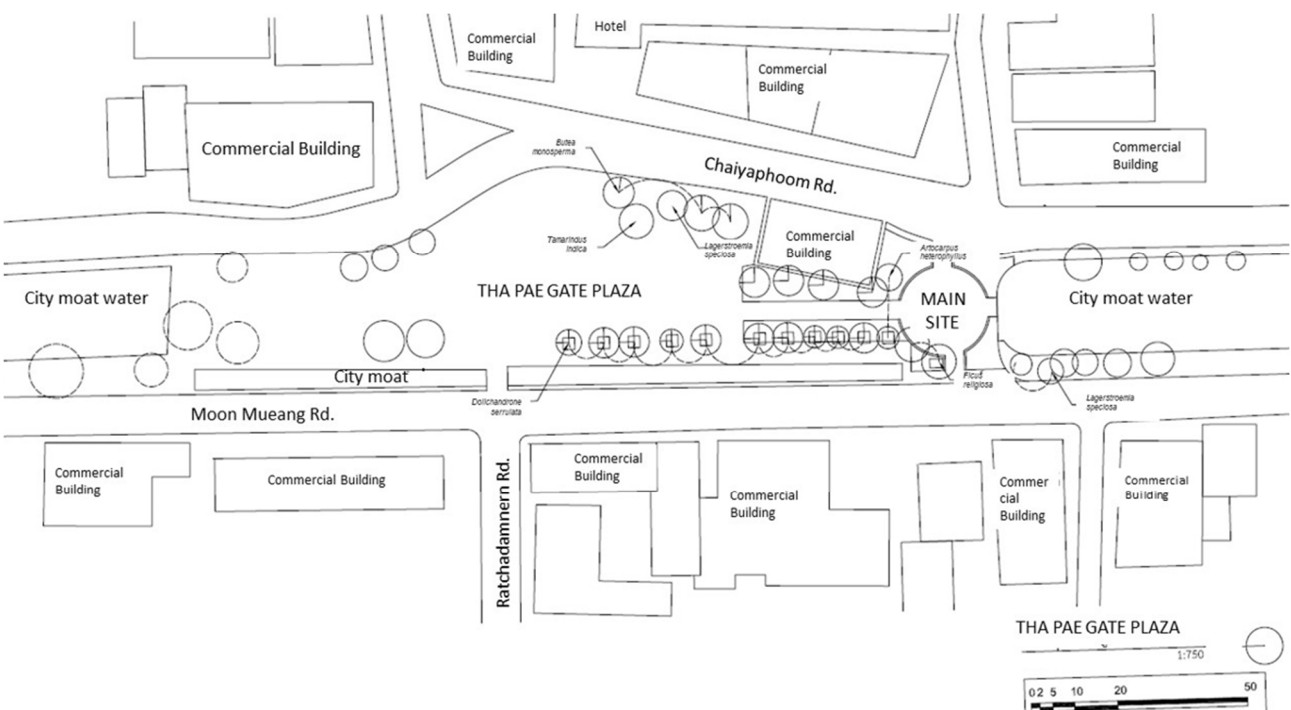

**Figure 5.** Base map of the site, including Tha Phae Gate on the north (**left hand side**) and the study site (**right hand side**).

### 2.3. Pilot Installation Conceptual Design Workshop

After identifying the site, the researchers invited eleven teams of landscape design experts and landscape architects to work on the design for the site. The teams were selected based on their expertise in the field and their interests. Six teams decided to participate in the two-day sketch design workshop to develop the water-sensitive design strategy for the pilot site.

The workshop was held in Chiang Mai during the rainy season of 2023. The event started with a site visit, a lecture about site history, and an update on water-sensitive urban design tools and knowledge. They then spent one afternoon and one morning developing their designs and presented the designs in the afternoon of the final day. One week after the workshop, the teams sent their design packages, which included plans, elevations, sketches, and conceptual explanations, to the researcher team. The research team offered to digitize some of the hand-drawn works to keep the data in digital format. During that time, a team of trained researchers recorded the design ideas and concepts in reflexive journals to include in the analysis of the developed designs.

The designs were then studied by two key researchers (CW and PS) and commented on by another researcher (DR) to identify possible themes that emerged from these experts and develop the lens through which WSUD could be considered during the design based on this pilot project.

### 2.4. Public Opinion Interviews

Following the workshop, we conducted public interviews at Tha Pae Gate. Participants, including bystanders, tourists, business owners, and others were invited on site for an in-depth interview, each lasting between 30 and 45 min. There, the participants were asked about their basic demographics. Then, they were asked about their ideas regarding Tha Phae Gate and their understanding of stormwater management and were shown the images of the designs to discuss their perceptions, acceptance, and preference. To ensure unbiased response recording, two researchers were assigned to each team, with each participant interacting solely with one researcher. Some participants were contacted

on site but chose to be interviewed at a later time in another public place. Researchers maintained reflexive journals and audio-recorded the interviews. Transcriptions of these recordings were subsequently prepared by the researchers who recorded the interviews.

The transcribed interviews and the reflexive journal, along with the debriefing with the interview teams, were used to develop thematic and sentiment analyses of the interviews. Furthermore, the transcripts were fed into a language processing model, Chat GPT 4.0, to perform a similar analysis. Both sets of analysis results were then triangulated with the existing literature about public perceptions of stormwater management to ensure the accuracy and validity of the data.

The interview protocol was approved by the Research Ethics Committee of Chiang Mai University under the code 2566/201.

## 3. Results

### 3.1. Workshop Results

Six initial prototypes of the pilot project were created. Each team had its own interpretations and strategies of the pilot project, ranging from cultural to commercial and from practical to futuristic. The concepts and explanation of the projects are described in Table 1, and the master plans of the designs are presented in Figure 6, while some highlights and diagrams can be found in Figure 7. The full design documentation can be found in the Supplementary Materials.

**Table 1.** Project descriptions.

| Team Number | Project Name | Conceptual Description |
| --- | --- | --- |
| 1 | Remake | A new tourist attraction that involves art on walls and canoeing activities along urban canals can accommodate many tourists. There is an underground parking area at the port gate because the original port gate was a tourist spot, but it did not have parking. Additionally, there is a drainage pipe made of natural materials, layered with topsoil, sand, and gravel. This acts as a water filter before it enters the urban canal. The filtering area can also be turned into a green space for the venue. |
| 2 | Water Retention Garden | Increases the usable area and green space. Uses native plants and spontaneous urban vegetation to treat rainwater. |
| 3 | Stormwater Management in Tha-Phae | The design involves transforming impermeable hard surfaces that do not allow water to seep through. There is an underground water storage tank, underground drainage pipes, and connections to public water pipes. Plants are cultivated in areas where water is discharged to allow them to absorb, filter the wastewater, and alleviate flooding issues. |
| 4 | Great Bowl and Green Yard (สลุงหลวงช่วงเขียว) | This design includes integrated learning for people to understand the rainwater treatment process. By creating visible basins (ponds), the steps of the process are showcased, from treatment to the use of vegetation for absorption, filtration, and flood mitigation. |
| 5 | Tank | The design is for a green space intended to slow down and absorb rainwater, using plants that are resistant to waterlogging and sunlight. An underground water tank is integrated to assist with drainage in flooded areas. Usable spaces within various areas of the location are also expanded. |
| 6 | Spiral Prototype | Consideration is given to filtering standing water or rainwater for daily use. There are two filtering methods. The first one uses various materials, including gravel, sand, and charcoal. The second method employs various plants that can absorb and utilize the water. |

Overall, the engagement of the designers addressed the concerns and issues of the site and the surrounding contexts while focusing on the stormwater management issues. These solutions could identify potential approaches to stormwater problems surrounding Chiang Mai City Moat and offer visual representations of what they might look like. Based on these six prototypes, four important themes emerged: the links to Tha Phae Gate, water retention and treatment, planting design, and educational and recreational functions.

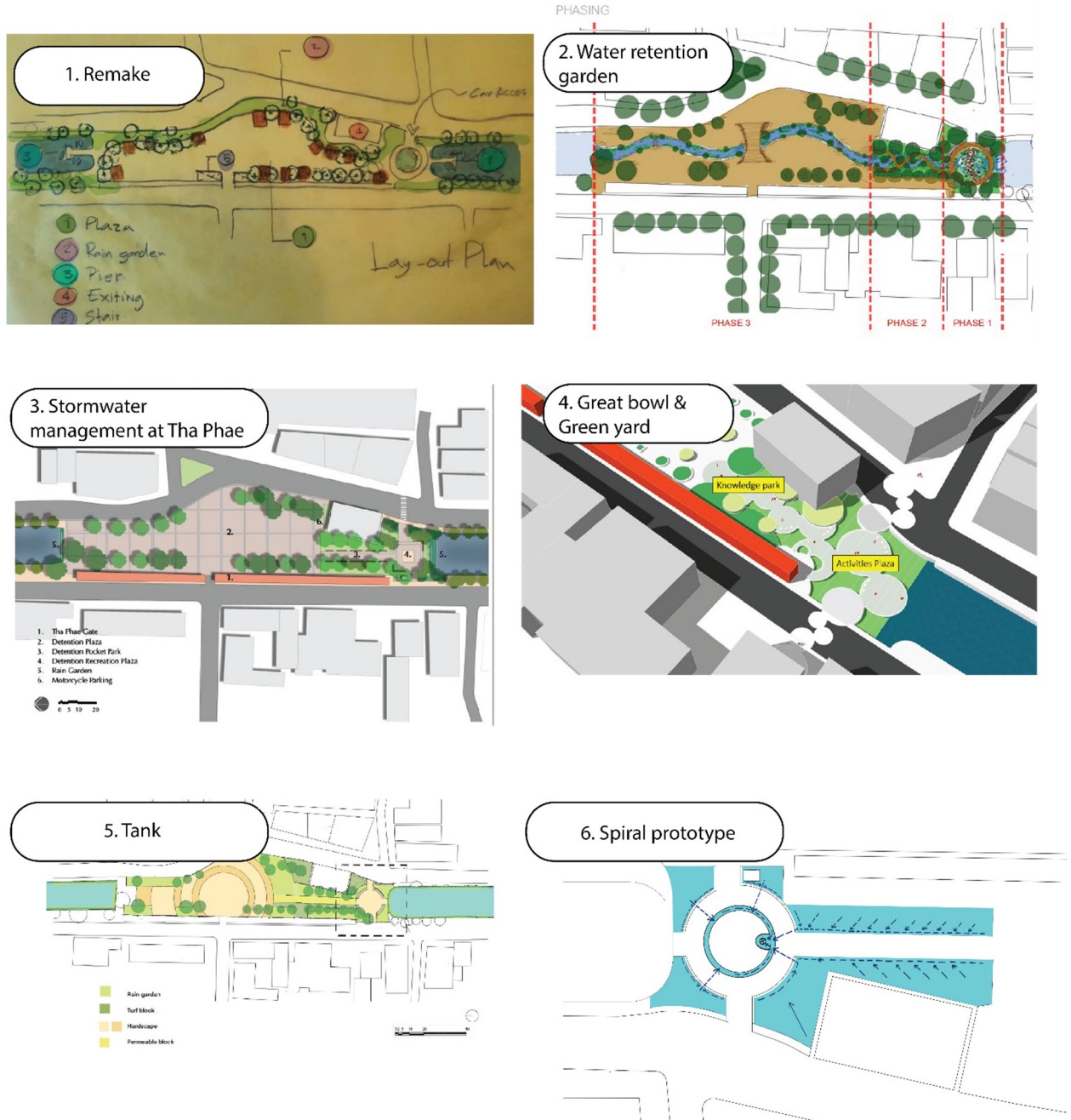

**Figure 6.** Design result masterplans.

### 3.1.1. Links to Tha-Phae Gate

The prototypes employed a range of involvement with the Tha-Phae Gate Plaza. Some of the designs, like the Tank and Spiral Prototype, focused on the site as a stand-alone component, leaving Tha-Phae Gate Plaza as it was, while some argued that the plaza and the surrounding canal provided much more potential both in terms of WSUD strategies and the social awareness of the project, such as the Remake, Water Retention Garden, and Stormwater Management in Tha-Phae prototypes. Another project, Great Bowl and Green Yard, fell in between the two approaches, using small design components to gently transition water and people from the plaza into the study site.

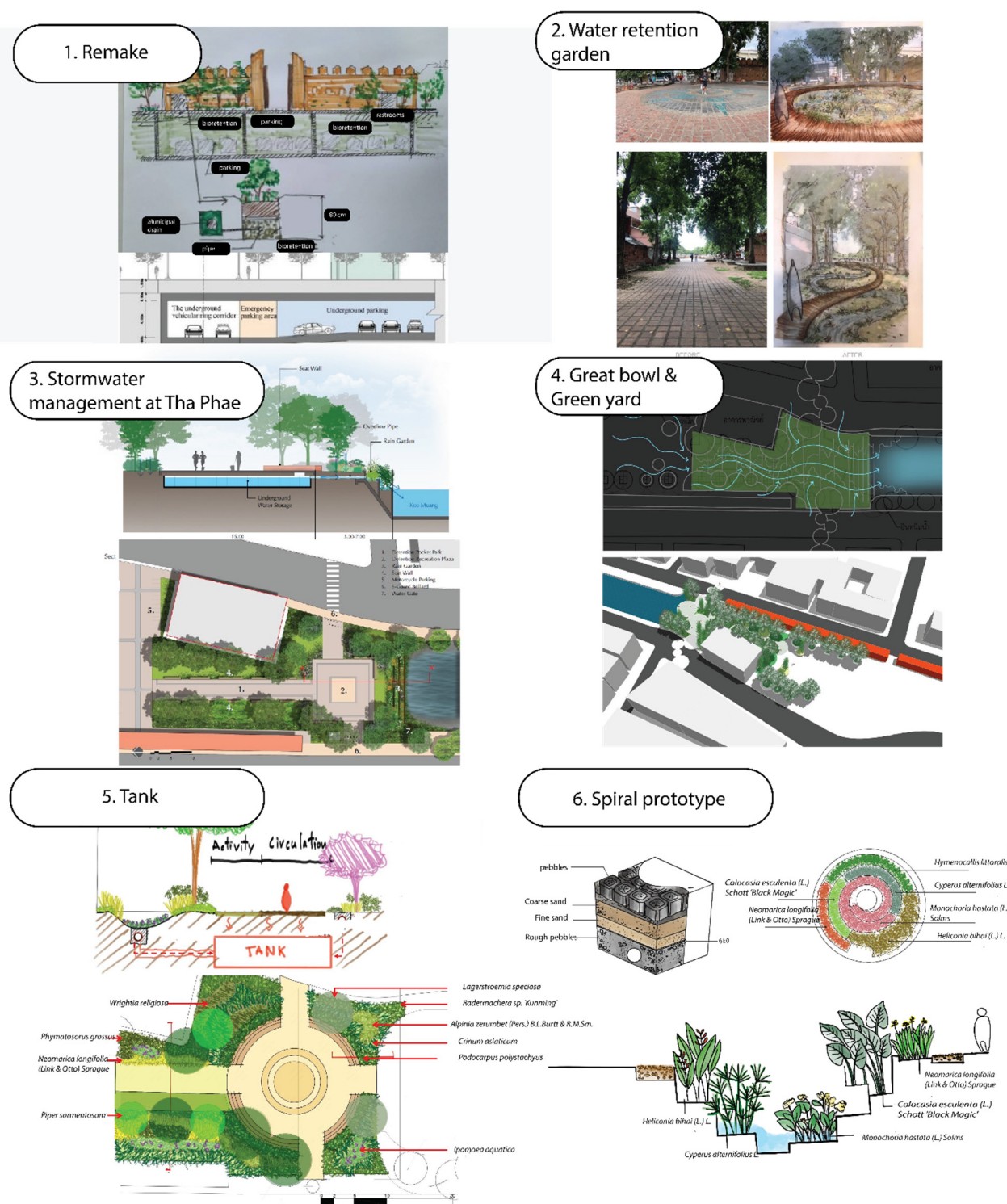

**Figure 7.** Highlight sketches and diagrams.

### 3.1.2. Water Retention and Treatments

When given a free choice of WSUD strategies, some designers opted for the strong hardscape solutions and focused on water retention, while others focused on softscapes and natural designs. The levels and management of water also differed between groups, ranging from storing gray water for landscaping (Stormwater Management in Tha-Phae), filtering water for releasing back to the moat (Water Retention Garden), to transforming some into drinking water (Spiral Prototype).

### 3.1.3. Planting Designs

Because of the limitations of space and the existing trees along the city moat, most designs focused on smaller trees, shrubs, and groundcovers that could help with phytoremediation. Some designs, such as Stormwater Management in Tha-Phae, focused on a larger plant community and the overall function of the space while others, such as Spiral Prototype, Water Retention Garden, and Tank, provided us with expansive planting plans that included local, amphibious, and spontaneous species based on their design themes.

### 3.1.4. Educational, Cultural, and Recreational Functions

Several designs highlighted aspects of education towards stormwater management, both in terms of water quantity and quality. Some designs highlighted the importance of plants (Tank and Water Retention Garden), while others focused on the water purification process (Spiral Prototype). Some used the site to address issues of the surrounding areas, such as parking, including Stormwater Management in Tha-Phae and Remake. Furthermore, Remake added layers of recreation, such as boating in the moat, while others (Great Bowl and Green Yard and Stormwater Management of Tha Phae) saw this as a cultural opportunity and added to the overall atmosphere to Tha Phae Gate Plaza as a whole. One design, Stormwater Management of Tha Phae, pulled inspiration from international tourist destinations like Cheonggyecheon to increase the tourism value of the site.

### *3.2. Interview Results*

Overall, 24 participants participated in interviews. Nineteen interviews were conducted on site, and five chose to be interviewed at Chiang Mai University later. Among the 24, 12 were male, and 12 were female. Seven participants identified themselves as young adults (20–30), twelve as adults (30–50), and six as older adults (50–65). Eleven were local, nine had moved and settled in Chiang Mai as their residence, and two were tourists.

Notably, eight participants worked in the area with various tourism-related jobs such as tourist photographer, delivery rider, masseuse, and rickshaw driver. The rest of the participants did not disclose their occupations, and thus, the researchers assumed they did not work within the site. Three participants directly disclosed to us that they used the area behind the shop as their overnight stay.

Among these participants, we have separated the themes of their conversations into four themes: stormwater management, experiences and activities, overall environment, and governance.

### 3.2.1. Stormwater Management

Many participants did not know nor think much about how stormwater should be managed. However, some did complain about the quality of the city moat water and its circulation. Many participants (10) also liked the idea of using plants and filters to improve stormwater quality. In terms of water quantity, more than half of the participants (13) agreed that underwater storage or more pervious surfaces may help address flooding and puddling around the space. Furthermore, some participants also added that they should be using the water features or the city moat water as a utility, such as drinking water or for raising fish and shellfish.

### 3.2.2. Experiences and Activities

All participants expressed their excitement about the experiences and activities, especially those related to tourism. One participant was concerned that any changes in design would change the current dynamic of the Tha Phae Gate Plaza, while others, especially those working in the areas, expressed their more urgent needs of facilities such as parking, seating, restrooms, and shade. A few participants thought that the place could be used as an educational space, but most think that something new and different may revitalize the space and the areas around the Tha Phae Gate Plaza.

### 3.2.3. Overall Environment and Nature

Most participants (17) expressed that increasing plants of different kinds would improve the beauty of the space. They also agreed in varying degrees that using nature-based solutions may add value to the place. Some also expressed an interest in other ecosystem services, such as thermal comfort, stress reduction, and air quality. One participant in particular discussed environmental sustainability and expressed doubt that one pilot project could contribute to any changes regarding the environmental movement in Chiang Mai. The perceived safety and cleanliness of the area were also discussed.

### 3.2.4. Governance

Interestingly, governance was not discussed at great length, but was a layered theme behind the interviews. Many potential participants rejected the invitation because they were afraid their participation could be used in a political campaign. When the governance was discussed (by six participants), it was about perceived safety and monitoring, maintenance, and budgeting. One participant, a homeless person, discussed that they feared the monitoring because they might get caught loitering after hours. Four participants discussed their feeling of powerlessness in making changes within the city or the environment.

When asked about where they thought stormwater management pilot projects should be, the participants recommended the entire city moat, Chiang Mai Gate, Warorot Market, Santhidhamma Region, Suandok Gate, and an abandoned green space next to Chiang Mai Train Station (Figure 8).

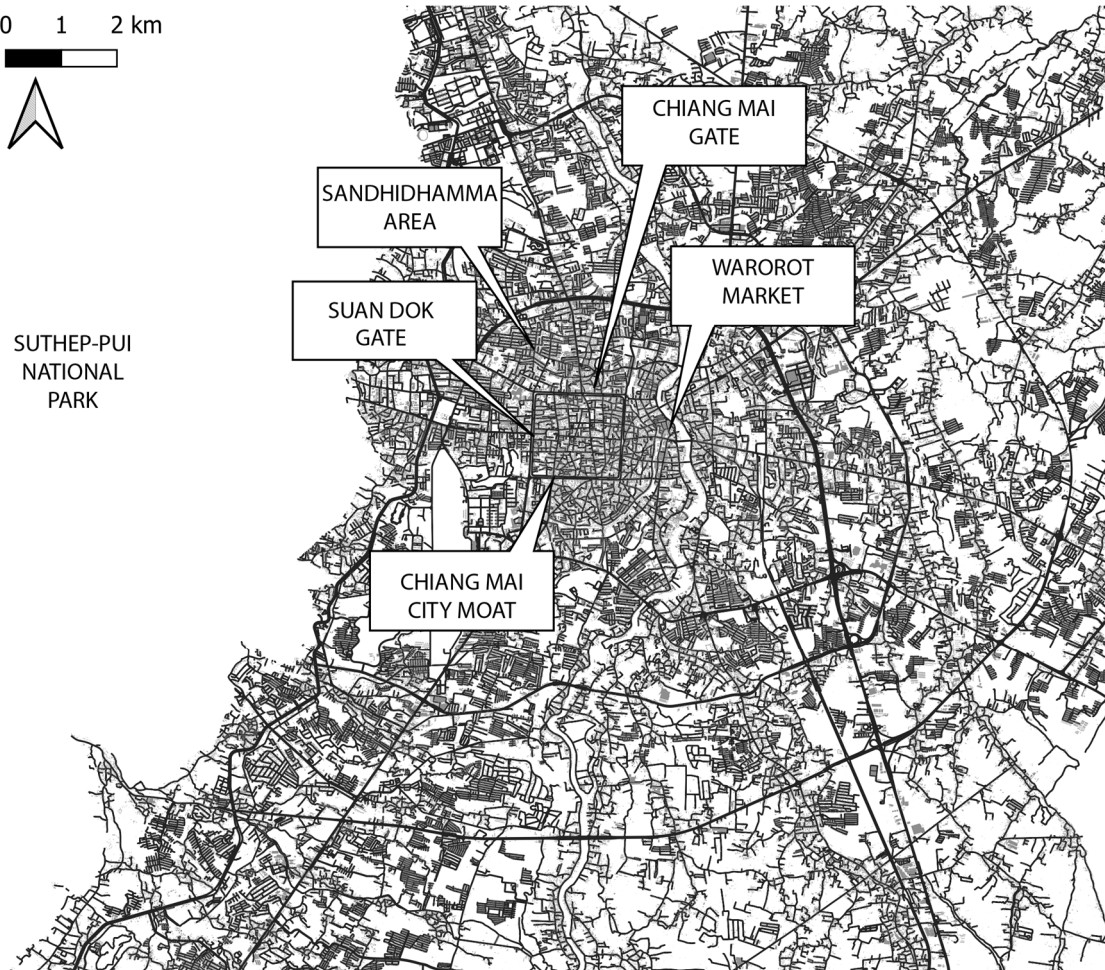

**Figure 8.** Suggested places for alternative pilot sites.

## 4. Discussion

### 4.1. Key Findings

In this study, the researchers aimed to address water quality issues in Chiang Mai, focusing on eutrophication from non-point source pollution by employing water-sensitive urban design (WSUD). They went through four phases, including site selection, a design workshop, on-site interviews, and data analysis. Key findings include the development of six design prototypes addressing stormwater management, with themes emerging around the linkage to Tha Phae Gate, water retention and treatment, planting designs, and educational, cultural, and recreational functions. The public interviews revealed varying levels of awareness and concerns regarding stormwater management, experiences and activities, environment and nature, and governance. These results could be used to understand the insights and priorities of designers and users.

### 4.2. Contributions

The information in this study added another layer to previous studies regarding the research and design relationship. Previously, researchers and designers have focused on different objectives regarding the design of public spaces [47,48], but based on the current design projects, the differences have lessened, as we see the designers leaning more toward sustainability [11,46,49,55,57,58]. This agreed with the New Landscape Declaration given by the Landscape Architecture Foundation in 2016 [46]. However, the differing focuses of the designers, researchers, and users remained. Many interviewees expressed ignorance or a dismissive attitude towards stormwater management and focused on cultural ecosystem services such as shade, restrooms, and tourism. This means that even though the research and design of WSUD are moving closer towards each other, the disconnection between designers' outcomes and the public remains. The designers for WSUD may need to communicate the functions and convey information more thoroughly than usual when engaged in projects like this one.

This knowledge adds another piece of evidence for the need to bridge the gaps between designers and laypeople, as identified in previous research [59,60]. This study contributes new insights towards WSUD by showcasing a possible range of design strategies through a collaborative workshop and gathering public opinions. Lastly, perceived safety, monitoring, and governance were lightly touched, similar to previous research [27,60,61], but due to people refusing to participate because they were afraid of a hidden political agenda, the results might be skewed by missing out on these participants. In a positive light, these insights include the employment of both hardscape and softscape solutions, exploring different water management techniques, and understanding the importance of public engagement and education in stormwater management initiatives.

In conclusion, the results of this study contributed to the research field by noting the focuses of designers and laypeople regarding WSUD elements. Furthermore, it identified the still existing gaps between researchers, designers, and laypeople—all with different priorities towards environmental design. Thus, the paper supplies another piece of evidence that collaboration between the three is needed in each project.

### 4.3. Implications and Recommendations

Designers and policy makers can use the process and findings from this study as a supplemental piece of evidence to inform WSUD initiatives both in Chiang Mai and globally. Even though the designs and sample sizes might not be strong enough on their own due to the limitations of the study, the generated alternatives and approaches could inspire designers and decision makers on how they could approach and pursue WSUD in their communities. The diverse design prototypes provide a set of strategies for managing stormwater. This emphasizes the need for public engagement, education, and a holistic approach that encompasses aesthetics, functionality, and cultural relevance. Moreover, insights into public perception and concerns can guide better communication and implementation strategies for WSUD projects, ensuring they are responsive to the

community's needs and preferences. Thus, designers and planners could focus on these factors during the planning, design, and development process of WSUD in the future.

*4.4. Limitations and Future Studies*

The main limitation of this study was the time used to produce the workshop. Due to the limited amount of time and stormwater data of Chiang Mai City Moat, the designers did not design for the hardscapes specification including soil characteristics, media layers, filtration systems, or any specific calculations for WSUD designs. This limitation made the performance rather conceptual in terms of stormwater quantity and quality. These designs should be further developed to include such factors, so that the environmental impacts can be calculated. Some other limitations might include a relatively small sample size in public interviews and potential biases in participant selection. The site-specific nature of the study may also limit the generalizability of the findings to other contexts. Future studies could benefit from a larger and more diverse participant pool, expanded geographical scope, and a comparative analysis with other WSUD projects. Additionally, engaging a wider range of stakeholders and employing more robust data analysis methods could further enrich the understanding and application of WSUD in different urban settings.

**5. Conclusions**

This investigation examined the potential of water-sensitive urban design (WSUD) in addressing water quality issues, specifically eutrophication, in Chiang Mai via collaborations between researchers, designers, and laypeople. Through a methodical approach encompassing site selection, design workshop, and public interviews, the study created design prototypes aimed at stormwater management. Key themes revolved around the integration with Tha Phae Gate, the balance between water retention and treatment, planting designs, and the combination of educational, cultural, and recreational functions. Public interviews unveiled data regarding perceptions and concerns regarding stormwater management and the experience, sustainability, and governance of the site, which are crucial for the successful implementation of WSUD strategies. Notably, the collaborative essence of the design workshop and the resonance of public opinions highlighted the significance of a community-centric approach in WSUD initiatives. However, the study also points towards certain limitations, including the limited time and data regarding soils and rainfall in Chiang Mai City Moat, which led to the limited ways to assess the stormwater performance of the designs, the scope of public engagement, and site-specific insights which may prompt a broader, more inclusive examination in future research. This exploration adds a practical dimension to the theoretical framework of WSUD and calls for more integrated, informed, and community-responsive urban water management practices in Chiang Mai and potentially other urban settings worldwide.

**Supplementary Materials:** The following supporting information can be downloaded at: https://www.mdpi.com/article/10.3390/su152216127/s1, Supplementary Document S1: Workshop Design Results.

**Author Contributions:** Conceptualization, P.S. and D.R.; methodology, P.S., N.C., V.S. and E.Y.; software, C.W.; validation, P.S. and C.W.; formal analysis, P.S.; investigation, P.S. and C.W.; resources, P.S.; data curation, P.S. and C.W.; writing—original draft preparation, P.S. and C.W.; writing—review and editing, P.S., N.C., V.S., C.W. and E.Y.; visualization, P.S.; supervision, D.R. All authors have read and agreed to the published version of the manuscript.

**Funding:** This project is funded by the Fundamental Fund Grant 2023, Chiang Mai University. The grant number is R65EX00154 under the project "Sustainable Climate-Urban-Water Resilience and Food Cycle Community Chiang Mai City Management".

**Institutional Review Board Statement:** The study was conducted in accordance with the Declaration of Helsinki and approved by the Research Ethics Committee of CHIANGMAI UNIVERSITY (protocol code 2566/201 at 28 August 2023).

**Informed Consent Statement:** Informed consent was obtained from all subjects involved in the study.

**Data Availability Statement:** Research data are available upon request to the corresponding author at pongsakorn.sup@cmu.ac.th.

**Acknowledgments:** The researchers would also like to thank City Research Design and Development Laboratory, Faculty of Engineering, CMU; 11.10 Landscape Studio; Wandee Architecture; 25 Ongsa Architecture; Axis Landscape; Landscape Design and Environmental Management Studio, Faculty of Agriculture, CMU; Kritanun Adirekkiet; Field and Hill; and SHMA-Soen for their active participation as designers in the project.

**Conflicts of Interest:** The authors declare no conflict of interest. The funders had no role in the design of the study; in the collection, analyses, or interpretation of data; in the writing of the manuscript; or in the decision to publish the results.

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
