# Peer review of "Enhancing Water-Sensitive Urban Design in Chiang Mai through a Research–Design Collaboration"

_sustainability, doi:10.3390/su152216127_

Round 1

Reviewer 1 Report

Comments and Suggestions for Authors

L27-28, L43. The following papers are recommended to provide context for urban expansion trends and associated risks within a global framework:

"Evaluating trends, benefits, and risks of global cities in recent urban expansion for advancing sustainable development. Habitat International…"

"Mapping global urban land for the 21st century with data-driven simulations and Shared Socioeconomic Pathways. Nat. Commun…"

In Fig.1, It is advisable to include remote sensing images to obtain a detailed understanding of the land use/cover in the study area.

The connection between research, design, and the public is not clearly delineated in the text.

In section 4.1, could you please provide some more meaningful findings, such as guidelines for developing water-sensitive cities, especially in the city or southeastern Asia. However, the role of each part remains somewhat unclear.

It is essential to clarify what can be learned from the study and the contribution it makes to the respective research field.

Comments on the Quality of English Language

Extensive editing of English language required.

Author Response

Enhancing Water-Sensitive Urban Design in Chiang Mai through Research-Design Collaboration

Responses to Reviewers

Dear reviewers and editorial office,

            Thank you for your time and input into the reviews of this manuscript. We highly appreciate your expertise and care into improving the quality of this paper into the publication standard of ‘Sustainability’.

            We have addressed the comments carefully and to the best of our abilities. These explanations of how the comments are addressed are included below.

Best,

Authors

Reviewer 1

L27-28, L43. The following papers are recommended to provide context for urban expansion trends and associated risks within a global framework:

"Evaluating trends, benefits, and risks of global cities in recent urban expansion for advancing sustainable development. Habitat International…"

"Mapping global urban land for the 21st century with data-driven simulations and Shared Socioeconomic Pathways. Nat. Commun…"

  • The citations have been added accordingly.

In Fig.1, It is advisable to include remote sensing images to obtain a detailed understanding of the land use/cover in the study area.

  • We assume that this refers to Fig. 2 since it shows the study area. While we did not have the most recent aerial photograph of Chiang Mai, we have added the land use regulation layer to give the land use and landcover in the study area.

The connection between research, design, and the public is not clearly delineated in the text.

  • The authors are unsure how to address this statement due to the many ways in which it could be interpreted. On the most direct note, we did mention the gaps between researchers, designers, and laypeople in section 1.4.
    • Aside from the lack of clear technical guides, there needs to be stronger connections between researchers, designers, and laypeople in landscape architecture, design, and planning. In 2000, a team of researchers interviewed a group of landscape architects and found that while the academia had shifted to focus towards sustainability, the practitioners opted to design mostly for human uses [47]. Similar studies in 2017 which analyzes the speech contents of practitioners showed a positive trend for more prominence of empirical evidence in landscape architecture and design [48]. However, there often are gaps expressed between researchers and designers in the field [49,50]. Traditionally, the practitioner for the researchers to produce the answers, and the re-searchers provided something that explains the questions rather than a practical solution [49,51]. Thus, more modern trends of landscape architecture, design, and planning research start focusing on the collaboration of the inquiries and start to blend to research and design together in various ways including studio-based research and re-search-through-design methods [49,50]. However, these solutions are under development and need examples and standards of how the processes might look like and what precaution may be needed from such collaboration.
  • Or maybe it was towards the interpretation of our results in section 4.2. If that was the case, we added the following sentence to clarify our statement.
    • This means that even though the research and design of WSUD are moving closer to-wards each other, the disconnection between designers’ outcome and the public remains.

In section 4.1, could you please provide some more meaningful findings, such as guidelines for developing water-sensitive cities, especially in the city or southeastern Asia. However, the role of each part remains somewhat unclear.

  • Section 4.1 focuses on the output—what we found directly from our study, while Section 4.3 provided the recommendations. However, we do agree that the texts could be made more clear. We have included the following sentences.
    • Section 4.1.: These results could be used to understand the insights and priorities of designers and users.
    • Section 4.3.: Thus, designers and planners could focus on these emphases during the planning, de-sign, and development process of WSUD in the future.

It is essential to clarify what can be learned from the study and the contribution it makes to the respective research field.

  • Section 4.2. was edited to clarify the contribution by adding the following paragraph.
    • In conclusion, the results of the study contributed to the research field by noting the focuses of designers and laypeople regarding WSUD elements. Furthermore, it identified the yet existing gaps between researchers, designers, and lay people—all with different priorities towards environmental design. Thus, the paper supplies another piece of evidence that collaborations between the three are needed in each project. 

Reviewer 2 Report

Comments and Suggestions for Authors

The paper focuses on the collaboration process between designers, researchers, and laypeople to create a Water Sensitive Urban Design (WSUD) prototype for Chiang Mai, such as a rain garden. It adds a practical dimension, more integrated and informed, to the theoretical framework of WSUD, calling a community-responsive urban water management practices. The problem is clearly stated, and solution proposals are welcomed.

Although the paper is well-intentioned, well-structured, and methodologically consistent, it has significant limitations. Almost all of them are identified by the authors. Concerning the site-specific nature, the authors recognize generalizability limitations to other contexts (lines 405-406) and, in addition, the small sample size and potential biases in public interviews can’t allow the authors to state “Designers and policymakers can use the findings from this study to inform WSUD initiatives […] globally” (lines 396-397). Please improve subsection 4.3.

However, the main limitation concerns hardscape (groundwater level and soil characteristics – horizon layers; porosity; permeability; consistency; compactness; shrinkage, plastic and liquid limits, etc.), which are forgotten in the designer team's proposals. In fact, the designs studied by two key researchers (CW and PS) (line 228), should be extended to the DR co-author (from the Department of Plant and Soil Sciences), in order to obtain more integrated and informed design proposals. This serious limitation doesn’t invalidate the well-intentioned paper, but it should be mentioned and discussed. Please improve sections 4 and 5.         

Minor improvements:

Lines 82-87: Please add worldwide examples, beyond the US examples of Figure 1

References: References [59-65] are missing in the References list

Author Response

Dear reviewers and editorial office,

            Thank you for your time and input into the reviews of this manuscript. We highly appreciate your expertise and care into improving the quality of this paper into the publication standard of ‘Sustainability’.

            We have addressed the comments carefully and to the best of our abilities. These explanations of how the comments are addressed are included below.

Best,

Authors

Reviewer 2

The paper focuses on the collaboration process between designers, researchers, and laypeople to create a Water Sensitive Urban Design (WSUD) prototype for Chiang Mai, such as a rain garden. It adds a practical dimension, more integrated and informed, to the theoretical framework of WSUD, calling a community-responsive urban water management practices. The problem is clearly stated, and solution proposals are welcomed.

  • The authors thank you for your kind comments and will uphold the standard of our strength in the future writing.

Although the paper is well-intentioned, well-structured, and methodologically consistent, it has significant limitations. Almost all of them are identified by the authors. Concerning the site-specific nature, the authors recognize generalizability limitations to other contexts (lines 405-406) and, in addition, the small sample size and potential biases in public interviews can’t allow the authors to state “Designers and policymakers can use the findings from this study to inform WSUD initiatives […] globally” (lines 396-397). Please improve subsection 4.3.

  • The sentence in subsection 4.3. has been changed to the following.
    • Designers and policymakers can use the process and findings from this study as a supplemental piece of evidence to inform WSUD initiatives both in Chiang Mai and globally. Even though the designs and sample sizes might not be strong enough on its own due to the limitations of the study, the generated alternative and approaches could inspire designers and decision makers how they could approach and pursue WSUD in their communities.

However, the main limitation concerns hardscape (groundwater level and soil characteristics – horizon layers; porosity; permeability; consistency; compactness; shrinkage, plastic and liquid limits, etc.), which are forgotten in the designer team's proposals. In fact, the designs studied by two key researchers (CW and PS) (line 228), should be extended to the DR co-author (from the Department of Plant and Soil Sciences), in order to obtain more integrated and informed design proposals. This serious limitation doesn’t invalidate the well-intentioned paper, but it should be mentioned and discussed. Please improve sections 4 and 5.         

  • For the line that was originally line 228, we edited the sentence as follow:
    • The designs were then studied by two key researchers (CW and PS) and commented by another researcher (DR), to identify possible themes that emerged from these experts and developed the lens of which WSUD could be considered during the design based on this pilot project.
  • Section 4, subsection 4.4, has been edited to include these sentences.
    • The main limitation of the study was the time used to produce the workshop. Due to the limited amount of time and stormwater data of Chiang Mai City Moat, the de-signers did not design for the hardscapes specification including soil characteristics, media layers, filtration systems, or any specific calculations for WSUD designs. This limitation made the performance towards stormwater quantity and quality rather conceptual. These designs should be further developed to include such factors so the environmental impacts can be calculated.
  • Section 5 has been edited to include this sentence.
    • However, the study also points towards certain limitations including the limited time and data regarding soils and rainfall in Chiang Mai City Moat, which led to the limited ways to assess stormwater performance of the design, the scope of public engagement and site-specific insights which may prompt a broader, more inclusive examination in future research.

Minor improvements:

Lines 82-87: Please add worldwide examples, beyond the US examples of Figure 1 

  • Examples from Australia have been added into the manuscript.

References: References [59-65] are missing in the References list

  • The citations are fixed. Now all the references have been presented in the work cited section.

Reviewer 3 Report

Comments and Suggestions for Authors

Review of the manuscript Enhancing Water-Sensitive Urban Design in Chiang Mai through Research-Design Collaboration.

 The authors in the article discussed the aspects related to the use of the Water Sensitive Urban Design (WSUD) in Chiang Mai City Moat. They present the results of the cooperation between researchers, designers, and laypeople in relation to the implementation of the WSUD solution in the part of the city.  In the reviewer's opinion, the topic is important, and the concept presented in the study shows that the integration of different practitioners leads to solutions that are more acceptable to society. The supplementary material is also interesting,

I have one objective related to the aspects associated with water quality – especially water quality problems. The authors in the abstract point out that “…study aimed to address water quality problems, especially eutrophication…”. In my opinion, this aspect is not presented in the article well. There are some general statements in the introduction section and result and discussion, but in the article body, this aspect is not presented appropriately. In my opinion, the way of using WSUD is the most valuable element of the paper. I suggest the author focus on this aspect and emphasize the role of cooperation between different groups of people involved in the WSUD. The issue related to eutrophication should be presented as a background of the study - potential results of the WSUD implementation. I recommend the authors slightly change the goals and emphasize the main results obtained from this study. 

 Generally, the article is well written and after minor changes the manuscript may be published.

Author Response

Dear reviewers and editorial office,

            Thank you for your time and input into the reviews of this manuscript. We highly appreciate your expertise and care into improving the quality of this paper into the publication standard of ‘Sustainability’.

            We have addressed the comments carefully and to the best of our abilities. These explanations of how the comments are addressed are included below.

Best,

Authors

The authors in the article discussed the aspects related to the use of the Water Sensitive Urban Design (WSUD) in Chiang Mai City Moat. They present the results of the cooperation between researchers, designers, and laypeople in relation to the implementation of the WSUD solution in the part of the city.  In the reviewer's opinion, the topic is important, and the concept presented in the study shows that the integration of different practitioners leads to solutions that are more acceptable to society. The supplementary material is also interesting,

  • The authors thank you for your kind comments and will uphold the standard of our strength in the future writing.

I have one objective related to the aspects associated with water quality – especially water quality problems. The authors in the abstract point out that “…study aimed to address water quality problems, especially eutrophication…”. In my opinion, this aspect is not presented in the article well. There are some general statements in the introduction section and result and discussion, but in the article body, this aspect is not presented appropriately. In my opinion, the way of using WSUD is the most valuable element of the paper. I suggest the author focus on this aspect and emphasize the role of cooperation between different groups of people involved in the WSUD. The issue related to eutrophication should be presented as a background of the study - potential results of the WSUD implementation. I recommend the authors slightly change the goals and emphasize the main results obtained from this study.

  • The aim of the study in the abstract has been changed as follows:
    • This study aimed to gather collaborations between researchers, designers, and laypeople to-wards WSUD, which has a potential to be implemented to address water quality issues.

Generally, the article is well written and after minor changes the manuscript may be published.

  • The authors thank you for your kind comments and will uphold the standard of our strength in the future writing.

Reviewer 4 Report

Comments and Suggestions for Authors

This paper reports on a collaborative process involving researchers, designers and the community informed by water sensitive urban design principles to address water quality issues in an urban setting in Thailand. The study involved site selection, a design workshop and onsite interviews in Chiang Mai City to create design prototypes aimed at stormwater management. The findings represent a contribution to a better understanding of WSUD practice in a tropical environment. This reviewer has not identified any substantial concerns regarding the topic, methodology or findings. The main issue which must be addressed before the paper can be published is English language editing.

Comments on the Quality of English Language

English language requires editing for grammar and syntax.

Author Response

Dear reviewers and editorial office,

            Thank you for your time and input into the reviews of this manuscript. We highly appreciate your expertise and care into improving the quality of this paper into the publication standard of ‘Sustainability’.

            We have addressed the comments carefully and to the best of our abilities. These explanations of how the comments are addressed are included below.

Best,

Authors

This paper reports on a collaborative process involving researchers, designers and the community. informed by water sensitive urban design principles to address water quality issues in an urban setting in Thailand. The study involved site selection, a design workshop and onsite interviews in Chiang Mai City to create design prototypes aimed at stormwater management. The findings represent a contribution to a better understanding of WSUD practice in a tropical environment. This reviewer has not identified any substantial concerns regarding the topic, methodology or findings. The main issue which must be addressed before the paper can be published is English language editing.

  • The authors thank you for your kind comments. The paper has been revised by a professional editor.

Reviewer 5 Report

Comments and Suggestions for Authors

Dear Authors, 

your article deals with an interesting and important issue. 

However, it seems that the research carried out on only one case is a bit small to be published in this journal. 

In the reviewer's opinion, the article should describe how the proposed solution can be implanted in other places. At the moment it is insufficiently described. 

In addition, the authors should indicate what is innovative about this research. 

The authors also did not indicate whether the proposed solution is also influenced by local regulations. 

In the reviewer's opinion, this research should be internationalized to make it suitable for publication. 

Author Response

Dear reviewers and editorial office,

            Thank you for your time and input into the reviews of this manuscript. We highly appreciate your expertise and care into improving the quality of this paper into the publication standard of ‘Sustainability’.

            We have addressed the comments carefully and to the best of our abilities. These explanations of how the comments are addressed are included below.

Best,

Authors

Dear Authors, your article deals with an interesting and important issue.

However, it seems that the research carried out on only one case is a bit small to be published in this journal. In the reviewer's opinion, the article should describe how the proposed solution can be implanted in other places. At the moment it is insufficiently described.

  • The Section 4 of the manuscript has been heavily edited to address the issue raised by the reviewer.

In addition, the authors should indicate what is innovative about this research. The authors also did not indicate whether the proposed solution is also influenced by local regulations.

  • The novelty and contribution of the research has been addressed in subsection 4.2 as follows.
    • In conclusion, the results of the study contributed to the research field by noting the focuses of designers and laypeople regarding WSUD elements. Furthermore, it identified the yet existing gaps between researchers, designers, and lay people—all with different priorities towards environmental design. Thus, the paper supplies an-other piece of evidence that collaboration between the three are needed in each project.

In the reviewer's opinion, this research should be internationalized to make it suitable for publication.

  • While we respect the reviewers’ stance on the generalizability of the project, especially for international audience, the authors would like to point out that many studies have done in a single country and have been published in this journal and other reputed journals—especially if the cases were studied in the EU countries [1,2], US [3], and Australia [4,5], and China[6]. Furthermore, having the Chiang Mai, Thailand context even provided novelty towards the study and have room to discuss how it may differ from the general, majorly ‘Western’ stance on the stormwater management approach.
  • We think that the paper is already up to the international standards due to its context and the pragmatic paradigm used in the study and are in equal quality with the papers published in this journal in this regard.
  1. Ristianti, N.S.; Bashit, N.; Ulfiana, D.; Windarto, Y.E. Bioretention Basin, Rain Garden, and Swales Track Concepts through Vegetated-WSUD: Sustainable Rural Stormwater Management in Klaten Regency. In Proceedings of the IOP Conference Series: Earth and Environmental Science, 2022; p. 012029.
  2. Rodrigues, M.; Antunes, C. Best management practices for the transition to a water-sensitive city in the south of Portugal. Sustainability 2021, 13, 2983.
  3. Suppakittpaisarn, P.; Chang, C.-Y.; Deal, B.; Larsen, L.; Sullivan, W.C. Does Vegetation Density and Perceptions Predict Green Stormwater Infrastructure Preference? Urban Forestry & Urban Greening 2020, 55, doi:https://doi.org/10.1016/j.ufug.2020.126842.
  4. Byrne, J.; Mouritz, M.; Taylor, M.; Breadsell, J.K. East village at Knutsford: A case study in sustainable urbanism. Sustainability 2020, 12, 6296.
  5. Byrne, J.; Taylor, M.; Wheeler, T.; Breadsell, J.K. WGV: Quantifying Mains Water Savings in a Medium Density Infill Residential Development. Sustainability 2020, 12, 6483.
  6. Sedrez, M.; Xie, J.; Cheshmehzangi, A. Integrating water sensitive design in the architectural design studio in china: Challenges and outcomes. Sustainability 2021, 13, 4853.

Round 2

Reviewer 1 Report

Comments and Suggestions for Authors

The paper has been well revised.

Comments on the Quality of English Language

 Minor editing of English language required

Reviewer 5 Report

Comments and Suggestions for Authors

Dear Authors, 

I do not quite agree with your answer about the internationalization of research, however, after corrections I think the article is suitable for publication.